# Meta-Learning with Adaptive Hyperparameters

**Sungyong Baik    Myungsub Choi    Janghoon Choi    Heewon Kim    Kyoung Mu Lee**
ASRI, Department of ECE, Seoul National University
{dsybaik, cms6539, ultio791, ghimhw, kyoungmu}@snu.ac.kr

## Abstract

The ability to quickly learn and generalize from only few examples is an essential goal of few-shot learning. Meta-learning algorithms effectively tackle the problem by learning how to learn novel tasks. In particular, model-agnostic meta-learning (MAML) encodes the prior knowledge into a trainable initialization, which allowed for fast adaptation to few examples. Despite its popularity, several recent works question the effectiveness of MAML initialization especially when test tasks are different from training tasks, thus suggesting various methodologies to improve the initialization. Instead of searching for a better initialization, we focus on a complementary factor in MAML framework, the inner-loop optimization (or fast adaptation). Consequently, we propose a new weight update rule that greatly enhances the fast adaptation process. Specifically, we introduce a small meta-network that can adaptively generate per-step hyperparameters: learning rate and weight decay coefficients. The experimental results validate that the **A**daptive **L**earning of hyperparameters for **F**ast **A**daptation (**ALFA**) is the equally important ingredient that was often neglected in the recent few-shot learning approaches. Surprisingly, fast adaptation from *random* initialization with **ALFA** can already outperform MAML.

## 1   Introduction

Inspired by the capability of humans to learn new tasks quickly from only few examples, few-shot learning tries to address the challenges of training artificial intelligence that can generalize well with the few samples. Meta-learning, or *learning-to-learn*, tackles this problem by investigating common prior knowledge from previous tasks that can facilitate rapid learning of new tasks. Especially, gradient (or optimization) based meta-learning algorithms are gaining increased attention, owing to its potential for generalization capability. This line of works attempts to directly modify the conventional optimization algorithms to enable fast adaptation with few examples.

One of the most successful instances for gradient-based methods is model-agnostic meta-learning (MAML) [8], where the meta-learner attempts to find a good starting location for the network parameters, from which new tasks are learned with few updates. Following this trend, many recent studies [3, 9, 11, 30, 39, 41] focused on learning a better initialization. However, research on the training strategy for fast adaptation to each task is relatively overlooked, typically resorting to conventional optimizers (*e.g.* SGD). Few recent approaches explore better learning algorithms for inner-loop optimization [6, 15, 26, 27], however they lack the adaptation property in weight updates, which is validated to be effective from commonly used adaptive optimizers, such as Adam [16].

In this paper, we turn our attention to an important but often neglected factor for MAML-based formulation of few-shot learning, which is the inner-loop optimization. Instead of trying to find a better initialization, we propose Adaptive Learning of hyperparameters for Fast Adaptation[1], named ALFA, that enables training to be more effective with task-conditioned inner-loop updates

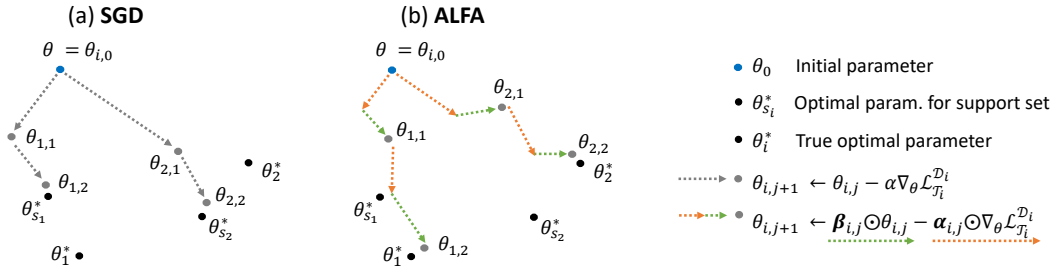

Figure 1: Overview of our proposed inner-loop optimization for few-shot learning. (a) Conventional optimizer (*e.g.*, SGD) updates the parameters in the direction of the gradient of task-specific loss $\nabla_{\boldsymbol{\theta}} \mathcal{L}_{\mathcal{T}_i}^{\mathcal{D}_i}$ with a fixed learning rate $\alpha$. The updates guide the parameters to the optimal values for training (or support) dataset, $\theta_{s_i}^*$. (b) ALFA *adapts* the learning rate $\boldsymbol{\alpha}_{i,j}$ and the regularization hyperparameter $\boldsymbol{\beta}_{i,j}$ w.r.t. the $i$-th task and the $j$-th inner-loop update step. The adaptive regularization effects of ALFA aims to facilitate better generalization to arbitrary unseen tasks, pushing the parameters closer to the true optimal values $\theta_i^*$.

from any given initialization. Our algorithm dynamically generates two important hyperparameters for optimization: learning rates and weight decay coefficients. Specifically, we introduce a small meta-network that generates these hyperparameters using the current weight and gradient values for each step, enabling each inner-loop iteration to be adaptive to the given task. As illustrated in Figure 1, ALFA can achieve better training and generalization, compared to conventional inner-loop optimization approaches, owing to per-step adaptive regularization and learning rates.

With the proposed training scheme ALFA, fast adaptation to each task from even a *random* initialization shows a better few-shot classification accuracy than MAML. This suggests that learning a good weight-update rule is at least as important as learning a good initialization. Furthermore, ALFA can be applied in conjunction with existing meta-learning approaches that aim to learn a good initialization.

## 2   Related work

The main goal of few-shot learning is to learn new tasks with given few support examples while maintaining the generalization to unseen query examples. Meta-learning aims to achieve the goal by learning prior knowledge from previous tasks, which in turn is used to quickly adapt to new tasks [4, 12, 32, 33, 36]. Recent meta-learning algorithms can be divided into three main categories: metric-based [17, 34, 35, 24, 38], network-based [21, 22, 31], and gradient-based [8, 23, 27, 28] algorithms. Among them, gradient (or optimization) based approaches are recently gaining increased attention for its potential for generalizability across different domains. This is because the gradient-based algorithms focus on adjusting the optimization algorithm itself, instead of learning feature space (metric-based) or designing network architecture specifically for fast adaptation (model-based). In this work, we concentrate on discussing gradient-based meta-learning approaches, in which there are two major directions: learning the initialization and learning the update rule.

One of the most recognized algorithms for learning a good initialization is MAML [8], which is widely used across diverse domains due to its simplicity and model-agnostic design. Such popularity led to a surge of MAML-based variants [2, 3, 9, 10, 11, 13, 14, 25, 26, 27, 30, 37, 39, 41, 42, 44], where they try to resolve the known issues of MAML, such as (meta-level) overfitting.

On the other hand, complementary studies on optimization algorithms, including better weight-update rules, have attracted relatively less attention from the community. This is evident from many recent MAML-based algorithms which settled with simple inner-loop update rules, without any regularization that may help prevent overfitting during fast adaptation to new tasks with few examples. Few recent works attempted to improve from such naïve update rules by meta-learning the learning rates [2, 20, 30] and learning to regularize the gradients [6, 10, 19, 25, 27]. However, these methods lack an adaptive property in the inner-loop optimization in that its meta-learned learning rate or regularization terms do not adapt to each task. By contrast, Ravi *et al.* [28] learn the entire inner-loop optimization directly through LSTM that generates updated weights (utilizing the design similar to [1]). While such formulation may be more general and provide a task-adaptive property, learning

the entire inner-loop optimization (especially generating weights itself) can be difficult and lacks the interpretability. This may explain why subsequent works, including MAML and its variants, resorted to simple weight-update rules (*e.g.*, SGD).

Therefore, we propose a new adaptive learning update rule for fast adaptation (ALFA) that is specifically designed for meta-learning frameworks. Notably, ALFA specifies the form of weight-update rule to include the learning rate and weight decay terms that are dynamically generated for each update step and task, through a meta-network that is conditioned on gradients and weights of a base learner. This novel formulation allows ALFA to strike a balance between learning of fixed learning rates for a simple weight-update rule (*e.g.*, SGD) [2, 20, 30] and direct learning of an entire complex weight-update rule [28].

## 3    Proposed method

### 3.1    Background

Before introducing our proposed method, we formulate a generic problem setting for meta-learning. Assuming a distribution of tasks denoted as $p(\mathcal{T})$, each task can be sampled from this distribution as $\mathcal{T}_i \sim p(\mathcal{T})$, where the goal of meta-learning is to learn prior knowledge from these sampled tasks. In a $k$-shot learning setting, $k$ number of examples $\mathcal{D}_i$ are sampled for a given task $\mathcal{T}_i$. After these examples are used to adapt a model to $\mathcal{T}_i$, a new set of examples $\mathcal{D}'_i$ are sampled from the same task $\mathcal{T}_i$ to evaluate the generalization performance of the adapted model on unseen examples with the corresponding loss function $\mathcal{L}_{\mathcal{T}_i}$. The feedback from the loss function $\mathcal{L}_{\mathcal{T}_i}$ is then used to adjust the model parameters to achieve higher generalization performance.

In MAML [8], the objective is to encode the prior knowledge from the sampled tasks into a set of common initial weight values $\boldsymbol{\theta}$ of the neural network $f_{\boldsymbol{\theta}}$, which can be used as a good initial point for fast adaptation to a new task. For a sampled task $\mathcal{T}_i$ with corresponding examples $\mathcal{D}_i$ and loss function $\mathcal{L}_{\mathcal{T}_i}^{\mathcal{D}_i}$, the network adapts to each task from its initial weights $\boldsymbol{\theta}$ with a fixed number of inner-loop updates. Network weights at time step $j$ denoted as $\boldsymbol{\theta}_{i,j}$ can be updated as:

$$\boldsymbol{\theta}_{i,j+1} = \boldsymbol{\theta}_{i,j} - \alpha \nabla_{\boldsymbol{\theta}} \mathcal{L}_{\mathcal{T}_i}^{\mathcal{D}_i}(f_{\boldsymbol{\theta}_{i,j}}), \tag{1}$$

where $\boldsymbol{\theta}_{i,0} = \boldsymbol{\theta}$. After $S$ number of inner-loop updates, task-adapted network weights $\boldsymbol{\theta}'_i = \boldsymbol{\theta}_{i,S}$ are obtained for each task. To evaluate and provide feedback for the generalization performance of the task-adapted network weights $\boldsymbol{\theta}'_i$, the network is evaluated with a new set of examples $\mathcal{D}'_i$ sampled from the original task $\mathcal{T}_i$. This outer-loop update acts as a feedback to update the initialization weights $\boldsymbol{\theta}$ to achieve better generalization across all tasks:

$$\boldsymbol{\theta} \leftarrow \boldsymbol{\theta} - \eta \nabla_{\boldsymbol{\theta}} \sum_{\mathcal{T}_i} \mathcal{L}_{\mathcal{T}_i}^{\mathcal{D}'_i}(f_{\boldsymbol{\theta}'_i}). \tag{2}$$

### 3.2    Adaptive learning of hyperparameters for fast adaptation (ALFA)

While previous MAML-based methods aim to find the common initialization weights shared across different tasks, our approach focuses on regulating the adaptation process itself through a learned update rule. To achieve the goal, we start by adding a $\ell_2$ regularization term $\frac{\lambda}{2}||\boldsymbol{\theta}||_2$ to the loss function $\mathcal{L}_{\mathcal{T}_i}$, which changes the inner-loop update equation (Equation (1)) as follows:

$$\begin{aligned} \boldsymbol{\theta}_{i,j+1} &= \boldsymbol{\theta}_{i,j} - \alpha(\nabla_{\boldsymbol{\theta}} \mathcal{L}_{\mathcal{T}_i}^{\mathcal{D}_i}(f_{\boldsymbol{\theta}_{i,j}}) + \lambda\boldsymbol{\theta}_{i,j}) \\ &= \beta\boldsymbol{\theta}_{i,j} - \alpha\nabla_{\boldsymbol{\theta}} \mathcal{L}_{\mathcal{T}_i}^{\mathcal{D}_i}(f_{\boldsymbol{\theta}_{i,j}}), \end{aligned} \tag{3}$$

where $\beta = 1 - \alpha\lambda$. We can control the adaptation process via the hyperparameters in the inner-loop update equation, which are scalar constants of learning rate $\alpha$ and regularization hyperparameter $\beta$. These hyperparameters can be replaced with adjustable variables $\boldsymbol{\alpha}_{i,j}$ and $\boldsymbol{\beta}_{i,j}$ with the same dimensions as $\nabla_{\boldsymbol{\theta}} \mathcal{L}_{\mathcal{T}_i}^{\mathcal{D}_i}(f_{\boldsymbol{\theta}_{i,j}})$ and $\boldsymbol{\theta}_{i,j}$, respectively. The final inner-loop update equation becomes:

$$\boldsymbol{\theta}_{i,j+1} = \boldsymbol{\beta}_{i,j} \odot \boldsymbol{\theta}_{i,j} - \boldsymbol{\alpha}_{i,j} \odot \nabla_{\boldsymbol{\theta}} \mathcal{L}_{\mathcal{T}_i}^{\mathcal{D}_i}(f_{\boldsymbol{\theta}_{i,j}}), \tag{4}$$

where $\odot$ denotes Hadamard (element-wise) product. To control the update rule for each task and each inner-loop update, we generate the hyperparameters based on the task-specific learning state

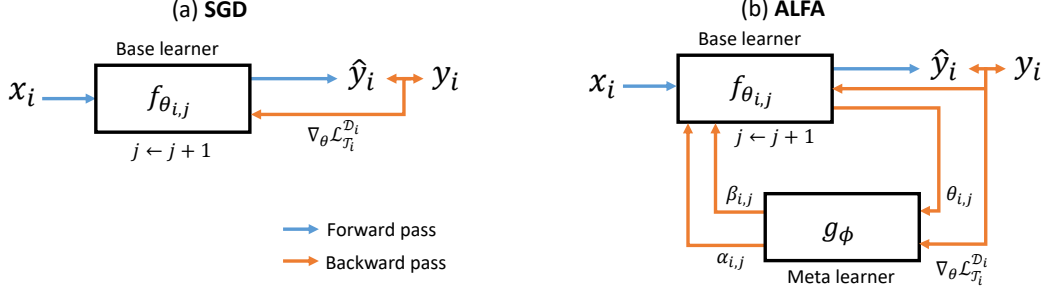

Figure 2: Illustration of the inner-loop update scheme. (a) Denoting the input, the output, and the label as $x_i$, $\hat{y}_i$, and $y_i$, respectively, conventional gradient-based meta-learning frameworks update the network parameters $\theta_{i,j}$ using simple update rules (*e.g.* SGD). (b) The proposed meta-learner $g_\phi$ generates the adaptive hyperparameters $\boldsymbol{\alpha}_{i,j}$ and $\boldsymbol{\beta}_{i,j}$ using the current parameters $\theta_{i,j}$ and its gradients $\nabla_{\boldsymbol{\theta}}\mathcal{L}_{\mathcal{T}_i}^{\mathcal{D}_i}$. Note that $\phi$ is only updated in the outer-loop optimization.

for task $\mathcal{T}_i$ at time step $j$, which can be defined as $\boldsymbol{\tau}_{i,j} = [\nabla_{\boldsymbol{\theta}}\mathcal{L}_{\mathcal{T}_i}^{\mathcal{D}_i}(f_{\boldsymbol{\theta}_{i,j}}), \boldsymbol{\theta}_{i,j}]$. We can generate hyperparameters $\boldsymbol{\alpha}_{i,j}$ and $\boldsymbol{\beta}_{i,j}$ from a neural network $g_\phi$ with network weights $\phi$ as follows:

$$(\boldsymbol{\alpha}_{i,j}, \boldsymbol{\beta}_{i,j}) = g_\phi(\boldsymbol{\tau}_{i,j}). \tag{5}$$

The hyperparameter generator network $g_\phi$ above produces the learning rate and regularization hyperparameters of weights in $\boldsymbol{\theta}_{i,j}$ for every inner-loop update step, in which they are used to control the direction and magnitude of the weight update. The overall process of the proposed inner-loop adaptation is depicted in Figure 2(b), compared to conventional approaches that use simple update rules, such as SGD, illustrated in Figure 2(a).

To train the network $g_\phi$, the outer-loop optimization using new examples $\mathcal{D}_i'$ and task-adapted weights $\boldsymbol{\theta}_i'$ is performed as in:

$$\phi \leftarrow \phi - \eta\nabla_\phi\sum_{\mathcal{T}_i}\mathcal{L}_{\mathcal{T}_i}^{\mathcal{D}_i'}(f_{\boldsymbol{\theta}_i'}). \tag{6}$$

Note that our method only learns the weights $\phi$ for the hyperparameter generator network $g_\phi$ while the initial weights $\boldsymbol{\theta}$ for $f_{\boldsymbol{\theta}}$ do not need to be updated throughout the training process. Therefore, our method can be trained to adapt from any given initialization (*e.g.*, random initializations). The overall training procedure is summarized in Algorithm 1. When using ALFA with MAML and its variants, the initialization parameter may be jointly trained to achieve higher performance.

Our adaptive inner-loop update rule bears some resemblance to gradient-descent based optimization algorithms [7, 16, 43], in which the learning rate of each weight can be regulated by the accumulated moments of past gradients. However, we propose a learning-based approach with the adaptive learning rate and the regularization hyperparameters that are generated by a meta-network that is explicitly trained to achieve generalization on unseen examples.

### 3.3 Architecture

For our proposed hyperparameter generator network $g_\phi$, we employ a 3-layer MLP with ReLU activation between the layers. For the computational efficiency, we reduce the task-specific learning state $\boldsymbol{\tau}_{i,j}$ to $\bar{\boldsymbol{\tau}}_{i,j}$, which are layer-wise means of gradients and weights, thus resulting in 2 state values per layer. Assuming a $N$-layer CNN for $f_{\boldsymbol{\theta}}$, the hyperparameter generator network $g_\phi$ takes $2N$-dimensional vector $\bar{\boldsymbol{\tau}}_{i,j}$ as input, with the same number of hidden units for intermediate layers. For outputs, the learning rate $\boldsymbol{\alpha}_{i,j}^1$ and the weight-decay term $\boldsymbol{\beta}_{i,j}^1$ are first generated layer-wise and then repeated to the dimensions of the respective parameters $\boldsymbol{\theta}_{i,j}$. Following the practices from [24], per-step per-layer meta-learnable post-multipliers are multiplied to the generated hyperparameter values to better control the range of the generated values for stable training. Mathematically, the learning rate and weight-decay terms are generated at the $j$-th step for the task $\mathcal{T}_i$ as in:

$$\begin{aligned} \boldsymbol{\alpha}_{i,j} &= \boldsymbol{\alpha}_{i,j}^0 \odot \boldsymbol{\alpha}_{i,j}^1(\bar{\boldsymbol{\tau}}_{i,j}), \\ \boldsymbol{\beta}_{i,j} &= \boldsymbol{\beta}_{i,j}^0 \odot \boldsymbol{\beta}_{i,j}^1(\bar{\boldsymbol{\tau}}_{i,j}), \end{aligned} \tag{7}$$

where $\boldsymbol{\alpha}_{i,j}^0, \boldsymbol{\beta}_{i,j}^0$ are meta-learnable post-multipliers and $\boldsymbol{\alpha}_{i,j}^1(\boldsymbol{\tau}_{i,j}), \boldsymbol{\beta}_{i,j}^1(\boldsymbol{\tau}_{i,j})$ are generated layer-wise multiplier values, all of which are repeated to the dimension of $\boldsymbol{\theta}_{i,j}$. Instead of generating the

---

**Algorithm 1** Adaptive Learning of Hyperparameters for Fast Adaptation (ALFA)

---

**Require:** Task distribution $p(\mathcal{T})$, learning rate $\eta$, arbitrary given initialization $\boldsymbol{\theta}$

1:  Randomly initialize $\boldsymbol{\phi}$
2:  **while** not converged **do**
3:      Sample a batch of tasks $\mathcal{T}_i \sim p(\mathcal{T})$
4:      **for** each task $\mathcal{T}_i$ **do**
5:          Initialize $\boldsymbol{\theta}_{i,0} = \boldsymbol{\theta}$
6:          Sample disjoint examples $(\mathcal{D}_i, \mathcal{D}'_i)$ from $\mathcal{T}_i$
7:          **for** inner-loop step $j := 0$ **to** $S - 1$ **do**
8:              Compute loss $\mathcal{L}_{\mathcal{T}_i}^{\mathcal{D}_i}(f_{\boldsymbol{\theta}_{i,j}})$ by evaluating $\mathcal{L}_{\mathcal{T}_i}$ w.r.t. $\mathcal{D}_i$
9:              Compute task-specific learning state $\boldsymbol{\tau}_{i,j} = [\nabla_{\boldsymbol{\theta}}\mathcal{L}_{\mathcal{T}_i}^{\mathcal{D}_i}(f_{\boldsymbol{\theta}_{i,j}}), \boldsymbol{\theta}_{i,j}]$
10:             Compute hyperparameters $(\boldsymbol{\alpha}_{i,j}, \boldsymbol{\beta}_{i,j}) = g_{\boldsymbol{\phi}}(\boldsymbol{\tau}_{i,j})$
11:             Perform gradient descent to compute adapted weights:
                $\boldsymbol{\theta}_{i,j+1} = \boldsymbol{\beta}_{i,j} \odot \boldsymbol{\theta}_{i,j} - \boldsymbol{\alpha}_{i,j} \odot \nabla_{\boldsymbol{\theta}}\mathcal{L}_{\mathcal{T}_i}^{\mathcal{D}_i}(f_{\boldsymbol{\theta}_{i,j}})$
12:         **end for**
13:         Compute $\mathcal{L}_{\mathcal{T}_i}^{\mathcal{D}'_i}(f_{\boldsymbol{\theta}'_i})$ by evaluating $\mathcal{L}_{\mathcal{T}_i}$ w.r.t. $\mathcal{D}'_i$ and task-adapted weights $\boldsymbol{\theta}'_i = \boldsymbol{\theta}_{i,S}$
14:     **end for**
15:     Perform gradient descent to update weights: $\boldsymbol{\phi} \leftarrow \boldsymbol{\phi} - \eta \nabla_{\boldsymbol{\phi}} \sum_{\mathcal{T}_i} \mathcal{L}_{\mathcal{T}_i}^{\mathcal{D}'_i}(f_{\boldsymbol{\theta}'_i})$
16: **end while**

---

hyperparameters $\boldsymbol{\alpha}_{i,j}$ and $\boldsymbol{\beta}_{i,j}$ for every element in $\boldsymbol{\theta}_{i,j}$ and $\nabla_{\boldsymbol{\theta}}\mathcal{L}_{\mathcal{T}_i}^{\mathcal{D}_i}(f_{\boldsymbol{\theta}_{i,j}})$, the layer-wise generation of hyperparameters makes our generator network $g_{\boldsymbol{\phi}}$ more computationally efficient, in which we can greatly reduce the number of weights that are trained during the outer-loop optimization. As for a random initialization, ALFA requires a meta-learnable per-parameter weight decay term to replace the role of MAML initialization in formulating the prior knowledge for each parameter of a base learner. In both cases, the overall number of learnable parameters increased from MAML, together with per-step per-layer post-multipliers, is minimal with $2SN + 12N^2$, where $S$ is the number of inner-loop steps and $N$ is the number of layers of a base learner $f_{\theta}$.

## 4 Experiments

In this section, we demonstrate the effectiveness of our proposed weight-update rule (ALFA) in few-shot learning. Even starting from a random initialization, ALFA can drive the parameter values closer to the optimal point than using a naïve SGD update for MAML initialization, suggesting that the inner-loop optimization is just as important as the outer-loop optimization.

### 4.1 Datasets

For few-shot classification, we use the two most popular datasets: miniImageNet [38] and tieredImageNet [29]. Both datasets are derived subsets of ILSVRC-12 dataset in specific ways to simulate the few-shot learning environment. Specifically, miniImageNet is composed of 100 classes randomly sampled from the ImageNet dataset, where each class has 600 images of size $84 \times 84$. To evaluate in few-shot classification settings, it is divided into 3 subsets of classes without overlap: 64 classes for meta-train set, 16 for meta-validation set, and 20 for meta-test set as in [28]. Similarly, tieredImageNet is composed of 608 classes with 779,165 images of size $84 \times 84$. The classes are grouped into 34 hierarchical categories, where 20 / 6 / 8 disjoint categories are used as meta-train / meta-validation / meta-test sets, respectively.

To take a step further in evaluating the rapid learning capability of meta-learning models, a cross-domain scenario is introduced in [5], in which the models are tested on tasks that are significantly different from training tasks. Specifically, we fix the training set to the meta-train set of miniImageNet and evaluate with the meta-test sets from CUB-200-2011 (denoted as CUB) [40].

Triantafillou *et al.* [37] recently introduced a large-scale dataset, named Meta-Dataset, which aims to simulate more realistic settings by collecting several datasets into one large dataset. Further challenges are introduced by varying the number of classes and the number of examples for each task and reserving two entire datasets for evaluation, similar to cross-domain settings where meta-train and meta-test sets differ.

Table 1: Test accuracy on 5-way classification for miniImageNet and tieredImageNet.

| | Backbone | miniImageNet | | tieredImageNet | |
|---|---|---|---|---|---|
| | | 1-shot | 5-shot | 1-shot | 5-shot |
| Random Init | 4-CONV | $24.85 \pm 0.43\%$ | $31.09 \pm 0.46\%$ | $26.55 \pm 0.44\%$ | $33.82 \pm 0.47\%$ |
| **ALFA** + Random Init | 4-CONV | $51.61 \pm 0.50\%$ | $70.00 \pm 0.46\%$ | $53.32 \pm 0.50\%$ | $71.97 \pm 0.44\%$ |
| MAML [8] | 4-CONV | $48.70 \pm 1.75\%$ | $63.11 \pm 0.91\%$ | $49.06 \pm 0.50\%$ | $67.48 \pm 0.47\%$ |
| **ALFA** + MAML | 4-CONV | $50.58 \pm 0.51\%$ | $69.12 \pm 0.47\%$ | $53.16 \pm 0.49\%$ | $70.54 \pm 0.46\%$ |
| MAML + L2F [3] | 4-CONV | $52.10 \pm 0.50\%$ | $69.38 \pm 0.46\%$ | $54.40 \pm 0.50\%$ | $73.34 \pm 0.44\%$ |
| **ALFA** + MAML + L2F | 4-CONV | $\mathbf{52.76 \pm 0.52}\%$ | $\mathbf{71.44 \pm 0.45}\%$ | $\mathbf{55.06 \pm 0.50}\%$ | $\mathbf{73.94 \pm 0.43}\%$ |
| Random Init | ResNet12 | $31.23 \pm 0.46\%$ | $41.60 \pm 0.49\%$ | $33.46 \pm 0.47\%$ | $44.54 \pm 0.50\%$ |
| **ALFA** + Random Init | ResNet12 | $56.86 \pm 0.50\%$ | $72.90 \pm 0.44\%$ | $62.00 \pm 0.47\%$ | $79.81 \pm 0.40\%$ |
| MAML | ResNet12 | $58.37 \pm 0.49\%$ | $69.76 \pm 0.46\%$ | $58.58 \pm 0.49\%$ | $71.24 \pm 0.43\%$ |
| **ALFA** + MAML | ResNet12 | $59.74 \pm 0.49\%$ | $\mathbf{77.96 \pm 0.41}\%$ | $\mathbf{64.62 \pm 0.49}\%$ | $\mathbf{82.48 \pm 0.38}\%$ |
| MAML + L2F | ResNet12 | $59.71 \pm 0.49\%$ | $77.04 \pm 0.42\%$ | $64.04 \pm 0.48\%$ | $81.13 \pm 0.39\%$ |
| **ALFA** + MAML + L2F | ResNet12 | $\mathbf{60.05 \pm 0.49}\%$ | $77.42 \pm 0.42\%$ | $64.43 \pm 0.49\%$ | $81.77 \pm 0.39\%$ |
| LEO-trainval [30] [*][†] | WRN-28-10 | $61.76 \pm 0.08\%$ | $77.59 \pm 0.12\%$ | $66.33 \pm 0.05\%$ | $81.44 \pm 0.09\%$ |
| MetaOpt [18] [*] | ResNet12 | $62.64 \pm 0.61\%$ | $78.63 \pm 0.46\%$ | $65.99 \pm 0.72\%$ | $81.56 \pm 0.53\%$ |

[*] Pre-trained network.
[†] Trained with a union of meta-training and meta-validation set.

## 4.2 Implementation details

For experiments with ImageNet-based datasets, we use 4-layer CNN (denoted as 4-CONV hereafter) and ResNet12 network architectures for the backbone feature extractor network $f_{\boldsymbol{\theta}}$. The 4-CONV and ResNet12 architectures used in this paper follow the same settings from [30, 34, 35, 38] and [24], respectively. In the meta-training stage, the meta-learner $g_{\phi}$ is trained over 100 epochs (each epoch with 500 iterations) with a batch size of 2 and 4 for 5-shot and 1-shot, respectively. At each iteration, we sample $N$ classes for $N$-way classification, followed by sampling $k$ labeled examples for $\mathcal{D}_i$ and 15 examples for $\mathcal{D}'_i$ for each class. In case for Meta-Dataset, all experiments were performed with the setup and hyperparameters provided by their source code [37][2]. For more details, please refer to the supplementary materials.

## 4.3 Experimental results

### 4.3.1 Few-shot classification

Table 1 summarizes the results of applying our proposed update rule ALFA on various initializations: random, MAML, and L2F (one of the state-of-the-art MAML-variant by Baik *et al.* [3]) on miniImageNet and tieredImageNet, along with comparisons to the other state-of-the-art meta-learning algorithms for few-shot learning. When the proposed update rule is applied on MAML, the performance is observed to improve substantially. What is even more interesting is that ALFA achieves high classification accuracy, even when applied on a random initialization, suggesting that meta-learning the inner-loop optimization (ALFA + Random Init) is more beneficial than solely meta-learning the initialization. This result underlines that the inner-loop optimization is as critical in MAML framework as the outer-loop optimization. We believe the promising results from ALFA can re-ignite focus and research on designing a better inner-loop optimzation, instead of solely focusing on improving the initialization (or outer-loop optimization). Table 1 further shows that our performance further improves when applied on MAML + L2F, especially for a small base learner backbone architecture (4-CONV). The fact that the proposed update rule can improve upon MAML-based algorithms proves the significance of designing a better inner-loop optimization. In addition, we present 20-way classification results for a 4-CONV base learner on miniImageNet in Table 4, which shows the significant performance boost after applying ALFA on MAML.

### 4.3.2 Cross-domain few-shot classification

To further justify the effectiveness of our proposed update rule in promoting fast adaptation, we perform experiments under cross-domain few-shot classification settings, where the meta-test tasks are substantially different from meta-train tasks. We report the results in Table 2, using the same

Table 2: Test accuracy on 5-way 5-shot cross-domain classification.

| | Backbone | miniImageNet $\rightarrow$ CUB |
|---|---|---|
| **ALFA** + Random Init | 4-CONV | $56.72 \pm 0.29\%$ |
| MAML [8] | 4-CONV | $52.70 \pm 0.32\%$ |
| **ALFA** + MAML | 4-CONV | $58.35 \pm 0.25\%$ |
| MAML + L2F [3] | 4-CONV | $60.89 \pm 0.22\%$ |
| **ALFA** + MAML + L2F | 4-CONV | $\mathbf{61.82 \pm 0.21}\%$ |
| **ALFA** + Random Init | ResNet12 | $60.13 \pm 0.23\%$ |
| MAML | ResNet12 | $53.83 \pm 0.32\%$ |
| **ALFA** + MAML | ResNet12 | $61.22 \pm 0.22\%$ |
| MAML + L2F | ResNet12 | $62.12 \pm 0.21\%$ |
| **ALFA** + MAML + L2F | ResNet12 | $\mathbf{63.24 \pm 0.22}\%$ |

experiment settings that are first introduced in [5], where miniImageNet is used as meta-train set and CUB dataset [40] as meta-test set.

The experimental results in Table 2 exhibit the similar tendency to few-shot classification results from Table 1. When a base learner with any initialization quickly adapts to a task from a new domain with ALFA, the performance is shown to improve significantly. The analysis in [5] suggests that a base learner with a deeper backbone is more robust to the intra-class variations in fine-grained classification, such as CUB. As the intra-class variation becomes less important, the difference between the support examples and query examples also becomes less critical, suggesting that the key lies in learning the support examples, without overfitting. This is especially the case when the domain gap between the meta-train and meta-test datasets is large, and the prior knowledge learned from the meta-training is mostly irrelevant. This makes learning tasks from new different domains difficult, as suggested in [3]. Thus, as discussed in [5], the adaptation to novel support examples plays a crucial role in cross-domain few-shot classification. Under such scenarios that demand the adaptation capability to new tasks, ALFA greatly improves the performance, further validating the effectiveness of the proposed weight update rule with adaptive hyperparameters.

### 4.3.3 Meta-Dataset

Table 3: Test accuracy on Meta-Dataset, where models are trained on ILSVRC-2012 only. Please refer to the supplementary materials for comparisons with state-of-the-art algorithms.

| | fo-MAML | | fo-Proto-MAML | |
|---|---|---|---|---|
| | | + ALFA | | + ALFA |
| ILSVRC | $45.51 \pm 1.11\%$ | $\mathbf{51.09 \pm 1.17}\%$ | $49.53 \pm 1.05\%$ | $\mathbf{52.80 \pm 1.11}\%$ |
| Omniglot | $55.55 \pm 1.54\%$ | $\mathbf{67.89 \pm 1.43}\%$ | $\mathbf{63.37 \pm 1.33}\%$ | $61.87 \pm 1.51\%$ |
| Aircraft | $56.24 \pm 1.11\%$ | $\mathbf{66.34 \pm 1.17}\%$ | $55.95 \pm 0.99\%$ | $\mathbf{63.43 \pm 1.10}\%$ |
| Birds | $63.61 \pm 1.06\%$ | $\mathbf{67.67 \pm 1.06}\%$ | $68.66 \pm 0.96\%$ | $\mathbf{69.75 \pm 1.05}\%$ |
| Textures | $\mathbf{68.04 \pm 0.81}\%$ | $65.34 \pm 0.95\%$ | $66.49 \pm 0.83\%$ | $\mathbf{70.78 \pm 0.88}\%$ |
| Quick Draw | $43.96 \pm 1.29\%$ | $\mathbf{60.53 \pm 1.13}\%$ | $51.52 \pm 1.00\%$ | $\mathbf{59.17 \pm 1.16}\%$ |
| Fungi | $32.10 \pm 1.10\%$ | $\mathbf{37.41 \pm 1.00}\%$ | $39.96 \pm 1.14\%$ | $\mathbf{41.49 \pm 1.17}\%$ |
| VGG Flower | $81.74 \pm 0.83\%$ | $\mathbf{84.28 \pm 0.97}\%$ | $\mathbf{87.15 \pm 0.69}\%$ | $85.96 \pm 0.77\%$ |
| Traffic Signs | $50.93 \pm 1.51\%$ | $\mathbf{60.86 \pm 1.43}\%$ | $48.83 \pm 1.09\%$ | $\mathbf{60.78 \pm 1.29}\%$ |
| MSCOCO | $35.30 \pm 1.23\%$ | $\mathbf{40.05 \pm 1.14}\%$ | $43.74 \pm 1.12\%$ | $\mathbf{48.11 \pm 1.14}\%$ |

We assess the effectiveness of ALFA on the large-scale and challenging dataset, Meta-Dataset. Table 3 presents the test accuracy of models trained on ImageNet (ILSVRC-2012) only, where the classification accuracy of each model (each column) is measured on each dataset meta-test test (each row). The table illustrates that ALFA brings the consistent improvement over fo-MAML (first-order MAML) and fo-Proto-MAML, which is proposed by Triantafillou *et al.* [37] to improve the MAML initialization at fc-layer. The consistent performance improvement brought by ALFA, even under such large-scale environment, further suggests the importance of the inner-loop optimization and the effectiveness of the proposed weight update rule.

| Table 4: 20-way classification | | |
|---|---|---|
| Model | 1-shot (%) | 5-shot (%) |
| MAML | 15.21±0.36 | 18.23±0.39 |
| ALFA+MAML | **22.03±0.41** | **35.33±0.48** |

| Table 5: Ablation study on $\tau$ | |
|---|---|
| Input | 5-shot (%) |
| weight only | 68.47±0.46 |
| gradient only | 67.98±0.47 |
| weight + gradient (ALFA) | **69.12±0.47** |

## 4.4 Ablation studies

In this section, we perform ablation studies to better analyze the effectiveness of ALFA, through experiments with 4-CONV as a backbone under 5-way 5-shot miniImageNet classification scenarios.

### 4.4.1 Controlling the level of adaptation

We start with analyzing the effect of hyperparamter adaptation by generating each hyperparameter individually for MAML and random initialization. To this end, each hyperparameter is either meta-learned (which is fixed after meta-training, similar to [20]) or generated (through our proposed network $g_\phi$) per step or per layer, as reported in Table 6. In general, making the hyperparameters adaptive improves the performance over fixed hyperparameters. Furthermore, controlling the hyperparameters differently at each layer and inner-loop step is observed to play a significant role in facilitating fast adaptation. The differences in the role of learning rate $\alpha$ and weight decay term $\beta$ can be also observed. In particular, the results indicate that regularization term plays a more important role than the learning rate for a random initialization. Regularization can be crucial for a random initialization, which can be susceptible to overfitting when trained with few examples.

Table 6: Effects of varying the adaptability for learning rate $\alpha$ and regularization term $\beta$. **fixed** or **adaptive** indicates whether the hyperparameter is meta-learned or generated by $g_\phi$, respectively.

| Initialization | | per step | per layer | **fixed** | **adaptive** |
|---|---|---|---|---|---|
| MAML | $\alpha$ | ✓ | | $64.76 \pm 0.48\%$ | $64.81 \pm 0.48\%$ |
| | | | ✓ | $64.52 \pm 0.48\%$ | $67.97 \pm 0.46\%$ |
| | $\beta$ | ✓ | | $66.78 \pm 0.45\%$ | $66.04 \pm 0.47\%$ |
| | | | ✓ | $66.30 \pm 0.47\%$ | $65.10 \pm 0.48\%$ |
| Random | $\alpha$ | ✓ | | $43.55 \pm 0.50\%$ | $44.00 \pm 0.50\%$ |
| | | | ✓ | $44.64 \pm 0.50\%$ | $46.62 \pm 0.50\%$ |
| | $\beta$ | ✓ | | $65.09 \pm 0.48\%$ | $67.06 \pm 0.47\%$ |
| | | | ✓ | $62.89 \pm 0.43\%$ | $66.35 \pm 0.47\%$ |

### 4.4.2 Inner-loop step

We make further analysis on the effectiveness of our method in fast adaptation by varying the number of update steps. Specifically, we measure the performance of ALFA+MAML when trained for a specified number of inner-loop steps and report the results in Table 7. Regardless of the number of steps, ALFA+MAML consistently outperforms MAML that performs fast adaptation with 5 steps.

Table 7: Varying the number of inner-loop steps for fast adaptation with ALFA+MAML.

| MAML | ALFA+MAML | | | | |
|---|---|---|---|---|---|
| step 5 | step 1 | step 2 | step 3 | step 4 | step 5 |
| $63.11 \pm 0.91\%$ | $69.48 \pm 0.46\%$ | $69.13 \pm 0.42\%$ | $68.67 \pm 0.43\%$ | $69.67 \pm 0.45\%$ | $69.12 \pm 0.47\%$ |

### 4.4.3 Ablation study on the learning state

To investigate the role of each part of the learning state (*i.e.*, base learner weights and gradients), we perform an ablation study, where only each part is solely fed into the meta-network, $g_\phi$. Table 5 summarizes the ablation study results. The meta-network conditioned on each part of the learning state still exhibits the performance improvement over MAML, suggesting that both parts play important roles. Our final model that is conditioned on both weights and gradients give the best performance, indicating that weights and gradients are complementary parts of the learning state.

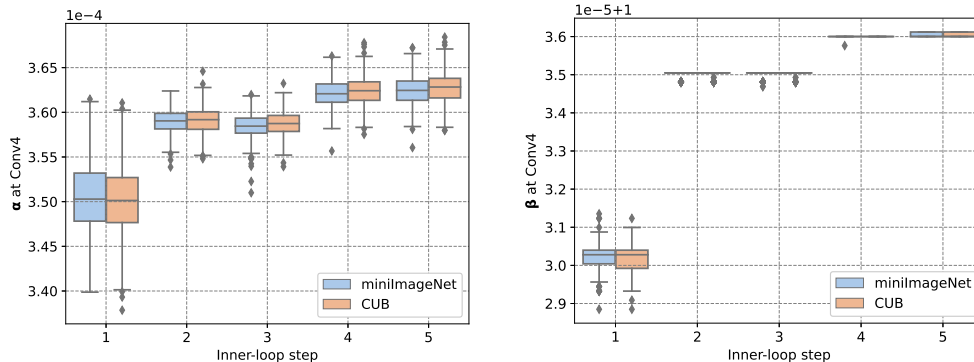

Figure 3: Visualization of the generated values of hyperparameters, $\alpha$ and $\beta$, across inner-loop steps for the 4-th convolutional layer. Dynamic ranges of generated values are also observed across different layers (please see the supplementary materials).

## 4.5 Few-shot regression

We study the generalizability of the proposed weight-update rule through experiments on few-shot regression. The objective of few-shot regression is to fit an unknown target function, given $k$ number of sampled points from the function. Following the settings from [8, 20], with the input range $[-5.0, 5.0]$, the target function is a sine curve with amplitude, frequency, and phase, which are sampled from intervals $[0.1, 5.0]$, $[0.8, 1.2]$, and $[0, \pi]$, respectively. We present results over $k = 5, 10, 20$ and a different number of network parameters in Table 8. ALFA consistently improves MAML, reinforcing the effectiveness and generalizability of the proposed weight-update rule.

Table 8: MSE over 100 sampled points with 95% confidence intervals on few-shot regression.

| Model | 2 hidden layers of 40 units | | | 3 hidden layers of 80 units | | |
|---|---|---|---|---|---|---|
| | 5 shots | 10 shots | 20 shots | 5 shots | 10 shots | 20 shots |
| MAML | 1.24±0.21 | 0.75±0.15 | 0.49±0.11 | 0.84±0.14 | 0.56±0.09 | 0.33±0.06 |
| ALFA+MAML | **0.92±0.19** | **0.62±0.16** | **0.34±0.07** | **0.70±0.15** | **0.51±0.10** | **0.25±0.06** |

## 4.6 Visualization of generated hyperparameters

We examine the hyperparameter values generated by ALFA to validate whether it actually exhibits the dynamic behaviour as intended. Through the visualization illustrated in Figure 3, we observe how the generated values differ for each inner-loop update step, under different domains (miniImageNet [38] and CUB [40]). We can see that the hyperparameters, learning rate $\alpha$ and regularization term $\beta$, are generated in a dynamic range for each inner-loop step. An interesting behavior to note is that the ranges of generated hyperparameter values are similar, under datasets from two significantly different domains. We believe such domain robustness is owed to conditioning on gradients and weights, which allow the model to focus on the correlation between generalization performance and the learning *trajectory* (weights and gradients), rather than domain-sensitive input image features.

## 5 Conclusion

We propose **ALFA**, an adaptive learning of hyperparameters for fast adaptation (or inner-loop optimization) in gradient-based meta-learning frameworks. By making the learning rate and the weight decay hyperparameters adaptive to the current learning state of a base learner, ALFA has been shown to consistently improve few-shot classification performance, regardless of initializations. Therefore, based on strong empirical validations, we claim that finding a good weight-update rule for fast adaptation is at least as important as finding a good initialization of the parameters. We believe that our results can initiate a number of interesting future research directions. For instance, one can explore different types of regularization methods, other than simple $\ell_2$ weight decay used in this work. Also, instead of conditioning only on layer-wise mean of gradients and weights, one can investigate other learning states, such as momentum.

## Broader Impact

Meta-learning and few-shot classification can help nonprofit organizations and small businesses automate their tasks at low cost, as only few labeled data may be needed. Due to the efficiency of automated tasks, nonprofit organizations can help more people from the minority groups, while small businesses can enhance their competitiveness in the world market. Thus, we believe that meta-learning, in a long run, will promote diversity and improve the quality of everyday life.

On the other hand, the automation may lead to social problems concerning job losses, and thus such technological improvements should be considered with extreme care. Better education of existing workers to encourage changing their roles (*e.g.*, managing the failure cases of intelligent systems, polishing the data for incremental learning) can help prevent unfortunate job losses.

## Acknowledgments and Disclosure of Funding

This work was supported by IITP grant funded by the Ministry of Science and ICT of Korea (No. 2017-0-01780); Hyundai Motor Group through HMG-SNU AI Consortium fund (No. 5264-20190101); and the HPC Support Project supported by the Ministry of Science and ICT and NIPA.

## Footnotes

[1]The code is available at `https://github.com/baiksung/ALFA`

[2]`https://github.com/google-research/meta-dataset`

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
