[Supplementary Material]

# Meta-Learning with Adaptive Hyperparameters
## – *Supplementary Document* –

In this supplementary document, we present the discussion on ResNet12 results (Section A) and additional results on few-shot classification (Section B) and cross-domain few-shot classification (Section D); experimental details (Section E); and more visualizations of generated hyperparameters (Section F).

## A    Discussion on ResNet12 results

Table A: 5-way 5-shot miniImageNet classification with multi-GPU setting vs single-GPU setting.

|  | ALFA+Random Init | MAML | ALFA+MAML | MAML+L2F | ALFA+MAML+L2F |
|---|---|---|---|---|---|
| Single-GPU [†] | $72.90 \pm 0.44\%$ | $69.76 \pm 0.46\%$ | $77.96 \pm 0.41\%$ | $77.04 \pm 0.42\%$ | $77.42 \pm 0.42\%$ |
| Multi-GPU | $88.90 \pm 0.31\%$ | $58.33 \pm 0.49\%$ | $88.36 \pm 0.32\%$ | $88.85 \pm 0.31\%$ | $90.92 \pm 0.29\%$ |

[†] The single GPU performance result is used in the main text.

We found a bug that is related to batch normalization in multi-GPU training/inference in the original MAML++ code [1], which our code is based on. The bug results in different performance depending on whether training/inference is performed with a single GPU or with multiple GPUs. We believe this is due to how (asynchronous) batch normalization behaves differently in a multi-GPU setting and MAML++ code does not shuffle the order of examples in a minibatch. This setting results in uneven class distribution across GPUs. While MAML performs worse in this setting, adaptive variants of MAML (L2F [2] or ALFA) perform substantially better, compared with a single-GPU setting (see Table A). This result suggests more investigation can be done on normalization in few-shot learning setting for possible performance improvement. While we report single-GPU ResNet12 results, we share our results and finding in hope of facilitating further research and study on the issue.

## B    Additional Experiments on Few-Shot Classification

We further validate the effectiveness of our proposed dynamic inner-loop update rule ALFA, through evaluating the performance on the relatively new CIFAR100-based [6] few-shot classification datasets: FC100 (Fewshot-CIFAR100) [12] and CIFAR-FS (CIFAR100 few-shots) [3]. They use low resolution images ($32 \times 32$) to create more challenging scenarios, compared to miniImageNet [14] and tieredImageNet [15], which use images of size $84 \times 84$. The difference between the two datasets comes from how CIFAR100 is split into meta-train / meta-validation / meta-test sets. Similar to tieredImageNet, FC100 splits the dataset based on superclasses, in order to minimize the amount of overlap. CIFAR-FS, on the other hand, is similar to miniImageNet, where the dataset is randomly split. Table B presents the results.

While ALFA with any initialization consistently performs better than MAML, the performance gap is not as significant as in miniImageNet, especially for a base learner with ResNet12 backbone. Also, unlike miniImageNet, ALFA with MAML+L2F does not always perform better than MAML+L2F. This may have to do with the low resolution of images, leading to noisy gradients. Gradients have more noise due to less data variations, compared to higher resolution of miniImageNet images. Because ALFA is mainly conditioned on the gradients, such noisy gradients are likely to disrupt

the generation of hyperparameters. While no data augmentation is used during training for fair comparisons with most meta-learning methods, data augmentation could help mitigate the problem as data augmentation may provide more data variations and thus less noisy gradients.

Table B: Test accuracy on 5-way classification for FC100 and CIFAR-FS.

| | Backbone | FC100 | | CIFAR-FS | |
|---|---|---|---|---|---|
| | | 1-shot | 5-shot | 1-shot | 5-shot |
| Random Init | 4-CONV | $27.50 \pm 0.45\%$ | $35.37 \pm 0.48\%$ | $29.74 \pm 0.46\%$ | $39.87 \pm 0.49\%$ |
| **ALFA** + Random Init | 4-CONV | $38.20 \pm 0.49\%$ | $52.98 \pm 0.50\%$ | **$60.56 \pm 0.49$**% | $75.43 \pm 0.43\%$ |
| MAML [†] [4] | 4-CONV | $36.67 \pm 0.48\%$ | $49.38 \pm 0.49\%$ | $56.80 \pm 0.49\%$ | $74.97 \pm 0.43\%$ |
| **ALFA** + MAML | 4-CONV | $37.99 \pm 0.48\%$ | $53.01 \pm 0.49\%$ | $59.96 \pm 0.49\%$ | **$76.79 \pm 0.42$**% |
| MAML + L2F [†] [2] | 4-CONV | **$38.96 \pm 0.49$**% | **$53.23 \pm 0.48$**% | $60.35 \pm 0.48\%$ | $76.76 \pm 0.42\%$ |
| **ALFA** + MAML + L2F | 4-CONV | $38.50 \pm 0.47\%$ | $53.20 \pm 0.50\%$ | $60.36 \pm 0.50\%$ | $76.60 \pm 0.42\%$ |
| Random Init | ResNet12 | $32.26 \pm 0.47\%$ | $42.00 \pm 0.49\%$ | $36.86 \pm 0.48\%$ | $49.46 \pm 0.50\%$ |
| **ALFA** + Random Init | ResNet12 | $40.57 \pm 0.49\%$ | $53.19 \pm 0.50\%$ | $64.14 \pm 0.48\%$ | $78.11 \pm 0.41\%$ |
| MAML [†] | ResNet12 | $37.92 \pm 0.48\%$ | $52.63 \pm 0.50\%$ | $64.33 \pm 0.48\%$ | $76.38 \pm 0.42\%$ |
| **ALFA** + MAML | ResNet12 | $41.46 \pm 0.49\%$ | **$55.82 \pm 0.50$**% | $66.79 \pm 0.47\%$ | **$83.62 \pm 0.37$**% |
| MAML + L2F [†] | ResNet12 | $41.89 \pm 0.47\%$ | $54.68 \pm 0.50\%$ | $67.48 \pm 0.46\%$ | $82.79 \pm 0.38\%$ |
| **ALFA** + MAML + L2F | ResNet12 | **$42.37 \pm 0.50$**% | $55.23 \pm 0.50\%$ | **$68.25 \pm 0.47$**% | $82.98 \pm 0.38\%$ |
| Prototypical Networks[*] [17] | 4-CONV | $35.3 \pm 0.6\%$ | $48.6 \pm 0.6\%$ | $55.5 \pm 0.7\%$ | $72.0 \pm 0.6\%$ |
| Relation Networks [18] | 4-CONV[+] | - | - | $55.0 \pm 1.0$ | $69.3 \pm 0.8$ |
| TADAM [12] | ResNet12 | $40.1 \pm 0.4\%$ | $56.1 \pm 0.4\%$ | - | - |
| MetaOpt [‡] [8] | ResNet12 | $41.1 \pm 0.6\%$ | $55.5 \pm 0.6\%$ | $72.0 \pm 0.7\%$ | $84.2 \pm 0.5\%$ |

[*] Meta-network is trained using the union of meta-training set and meta-validation set.
[+] Number of channels for each layer is modified to 64-96-128-256 instead of the standard 64-64-64-64.
[†] Our reproduction.
[‡] Meta-network is trained with data augmentation.

In Table C, we also add more comparisons to the prior works for Table 1 of our main paper, which were omitted due to space limit.

Table C: Test accuracy on 5-way classification for miniImageNet and tieredImageNet.

| | Backbone | miniImageNet | | tieredImageNet | |
|---|---|---|---|---|---|
| | | 1-shot | 5-shot | 1-shot | 5-shot |
| Random Init | 4-CONV | $24.85 \pm 0.43\%$ | $31.09 \pm 0.46\%$ | $26.55 \pm 0.44\%$ | $33.82 \pm 0.47\%$ |
| **ALFA** + Random Init | 4-CONV | $51.61 \pm 0.50\%$ | $70.00 \pm 0.46\%$ | $53.32 \pm 0.50\%$ | $71.97 \pm 0.44\%$ |
| MAML [4] | 4-CONV | $48.70 \pm 1.75\%$ | $63.11 \pm 0.91\%$ | $49.06 \pm 0.50\%$ | $67.48 \pm 0.47\%$ |
| **ALFA** + MAML | 4-CONV | $50.58 \pm 0.51\%$ | $69.12 \pm 0.47\%$ | $53.16 \pm 0.49\%$ | $70.54 \pm 0.46\%$ |
| MAML + L2F [2] | 4-CONV | $52.10 \pm 0.50\%$ | $69.38 \pm 0.46\%$ | $54.40 \pm 0.50\%$ | $73.34 \pm 0.44\%$ |
| **ALFA** + MAML + L2F | 4-CONV | **$52.76 \pm 0.52$**% | **$71.44 \pm 0.45$**% | **$55.06 \pm 0.50$**% | **$73.94 \pm 0.43$**% |
| Random Init | ResNet12 | $31.23 \pm 0.46\%$ | $41.60 \pm 0.49\%$ | $33.46 \pm 0.47\%$ | $44.54 \pm 0.50\%$ |
| **ALFA** + Random Init | ResNet12 | $56.86 \pm 0.50\%$ | $72.90 \pm 0.44\%$ | $62.00 \pm 0.47\%$ | $79.81 \pm 0.40\%$ |
| MAML | ResNet12 | $58.37 \pm 0.49\%$ | $69.76 \pm 0.46\%$ | $58.58 \pm 0.49\%$ | $71.24 \pm 0.43\%$ |
| **ALFA** + MAML | ResNet12 | $59.74 \pm 0.49\%$ | **$77.96 \pm 0.41$**% | $64.62 \pm 0.49\%$ | **$82.48 \pm 0.38$**% |
| MAML + L2F | ResNet12 | $59.71 \pm 0.49\%$ | $77.04 \pm 0.42\%$ | $64.04 \pm 0.48\%$ | $81.13 \pm 0.39\%$ |
| **ALFA** + MAML + L2F | ResNet12 | **$60.05 \pm 0.49$**% | $77.42 \pm 0.42\%$ | $64.43 \pm 0.49\%$ | $81.77 \pm 0.39\%$ |
| Matching Networks [20] | 4-CONV | $43.56 \pm 0.84\%$ | $55.31 \pm 0.73\%$ | - | - |
| Meta-Learning LSTM [14] | 4-CONV | $43.44 \pm 0.77\%$ | $60.60 \pm 0.71\%$ | - | - |
| Prototypical Networks[*] [17] | 4-CONV | $49.42 \pm 0.78\%$ | $68.20 \pm 0.66\%$ | $53.31 \pm 0.89\%$ | $72.69 \pm 0.74\%$ |
| Relation Networks [18] | 4-CONV[+] | $50.44 \pm 0.82\%$ | $65.32 \pm 0.70\%$ | $54.48 \pm 0.93\%$ | $71.32 \pm 0.78\%$ |
| Transductive Prop Nets [9] | 4-CONV | $55.51 \pm 0.99\%$ | $68.88 \pm 0.92\%$ | $59.91 \pm 0.94\%$ | $73.30 \pm 0.75\%$ |
| SNAIL [10] | ResNet12 | $55.71 \pm 0.99\%$ | $68.88 \pm 0.92\%$ | - | - |
| AdaResNet [11] | ResNet12 | $56.88 \pm 0.62\%$ | $71.94 \pm 0.57\%$ | - | - |
| TADAM [12] | ResNet12 | $58.50 \pm 0.30\%$ | $76.70 \pm 0.30\%$ | - | - |
| Activation to Parameter[*] [13] | WRN-28-10 | $59.60 \pm 0.41\%$ | $73.74 \pm 0.19\%$ | - | - |
| LEO-trainval[*] [16] | WRN-28-10 | $61.76 \pm 0.08\%$ | $77.59 \pm 0.12\%$ | $66.33 \pm 0.05\%$ | $81.44 \pm 0.09\%$ |
| MetaOpt [‡] [8] | ResNet12 | $62.64 \pm 0.61\%$ | $78.63 \pm 0.46\%$ | $65.99 \pm 0.72\%$ | $81.56 \pm 0.53\%$ |

[*] Meta-network is trained using the union of meta-training set and meta-validation set.
[+] Number of channels for each layer is modified to 64-96-128-256 instead of the standard 64-64-64-64.
[‡] Meta-network is trained with data augmentation.

## C    Comparisons with the state-of-the-art on Meta-Dataset

We compare one of the methods [19] that provides the state-of-the-art performance on Meta-Dataset. The state-of-the-art method is shown to outperform ALFA+fo-Proto-MAML in Table D. This is mainly because Tian *et al.* [19] uses the metric-based meta-learning approaches, which are known for high performance in few-shot classification. On the other hand, ALFA is a general plug-in module that can be used to improve over MAML-based algorithms, such as fo-Proto-MAML, as shown in Table 3 in the main paper. Also, ALFA can be used to improve over MAML-based algorithms in other problem domains, such as regression (shown in Table 8 of the main paper), while the algorithm from [19] can only be applied to few-shot classification.

Table D: Test accuracy on Meta-Dataset, where models are trained on ILSVRC-2012 only.

|  | ALFA+fo-Proto-MAML | Best from [19] |
|---|---|---|
| ILSVRC | 52.80% | 61.48% |
| Omniglot | 61.87% | 64.31% |
| Aircraft | 63.43% | 62.32% |
| Birds | 69.75% | 79.47% |
| Textures | 70.78% | 79.28% |
| Quick Draw | 59.17% | 60.84% |
| Fungi | 41.49% | 48.53% |
| VGG Flower | 85.96% | 91.00% |
| Traffic Signs | 60.78% | 76.33% |
| MSCOCO | 48.11% | 59.28% |

## D    Additional Experiments on Cross-Domain Few-Shot Classification

In this section, we study how robust the proposed meta-learner is to changes in domains, through additional experiments on cross-domain few-shot classification under similar settings to Section 4.3.2 in the main paper. In particular, miniImagenet meta-train set is used for meta-training, while corresponding meta-test splits of Omniglot [7], FC100 [12], and CIFAR-FS [3] are used for evaluation. Because either image channel (1 for Omniglot) or resolution ($28 \times 28$ for Omniglot and $32 \times 32$ for CIFAR-based datasets) is different from miniImagenet, we expand the image channel (to 3) and resolution (to $84 \times 84$) to match meta-train settings. Table E reports the test accuracy on 5-way 5-shot cross-domain classification of 4-CONV base learner with baseline meta-learners and our proposed meta-learners. Trends similar to Table 2 in the main paper are observed in Table E, where ALFA consistently improves the performance across different domains.

Table E: Test accuracy on 5-way 5-shot cross-domain classification. All models are only trained with miniImageNet meta-train set and tested on various datasets (domains) without any fine-tuning.

|  | miniImageNet | | |
|---|---|---|---|
|  | $\rightarrow$ **Omniglot** | $\rightarrow$ **FC100** | $\rightarrow$ **CIFAR-FS** |
| **ALFA** + Random Init | $91.02 \pm 0.29\%$ | $62.49 \pm 0.48\%$ | $63.49 \pm 0.45\%$ |
| MAML [4] | $85.68 \pm 0.35\%$ | $55.52 \pm 0.50\%$ | $55.82 \pm 0.50\%$ |
| **ALFA** + MAML | $93.11 \pm 0.23\%$ | $60.12 \pm 0.49\%$ | $59.76 \pm 0.49\%$ |
| MAML + L2F [2] | $94.96 \pm 0.22\%$ | $61.99 \pm 0.49\%$ | $63.73 \pm 0.48\%$ |
| **ALFA** + MAML + L2F | $94.10 \pm 0.24\%$ | $63.33 \pm 0.45\%$ | $63.87 \pm 0.48\%$ |

## E    Experimental Details

For the better reproducibility, the details of experiment setup, training, and architecture are delineated.

### E.1    Experiment Setup

For $N$-way $k$-shot classification on all datasets, the standard settings [4] are used. During the fast adaptation (inner-loop optimization), the number of examples in $\mathcal{D}$ is $k$ per each class. Except for the

ablation studies on the number of inner-loop steps (Section 4.4.2 in the main paper), the inner-loop optimization is performed for 5 gradient steps for all experiments performed in this work. During outer-loop optimization, 15 examples are sampled per each class for $\mathcal{D}'$. All models were trained for 50000 iterations with the meta-batch size of 2 and 4 tasks for 5-shot and 1-shot, respectively. For fair comparisons with most meta-learning methods, no data augmentation is used. Similar to the experimental settings from [1], an ensemble of the top 5 performing per-epoch-models on the validation set were evaluated on the test set. Every result is presented with the mean and standard deviation after running experiments independently with 3 different random seeds. All experiments were performed on NVIDIA GeForce GTX 2080Ti GPUs. For new experiments on ResNet12 backbone, NVIDIA Quadro RTX 8000 GPUs are used.

### E.2 Network Architecture for base learner $f_\theta$

**4-CONV** Following the settings from [1], 4 layers of 48-channel $3 \times 3$ convolution filters, batch normalization [5], Leaky ReLU non-linear activation functions, and $2 \times 2$ max pooling are used to build 4-CONV base learner. Then, the fully-connected layer and softmax are placed at the end of the base learner network.

**ResNet12** For overall ResNet12 architecture design, the settings from [12] are used. Specifically, the network is comprised of 4 residual blocks, each of which in turn consists of three convolution blocks. The first two convolution blocks in each residual block consist of $3 \times 3$ convolutional layer, batch normalization, and a ReLU non-linear activation function. In the last convolution block in each residual block, the convolutional layer is followed by batch normalization and a skip connection. Each skip connection contains a $1 \times 1$ convolutional layer, which is followed by batch normalization. Then, a ReLU non-linear activation function and $2 \times 2$ max-pooling are placed at the end of each residual block. Lastly, the number of filters is 64, 128, 256, 512 for each residual block, respectively.

### E.3 Network Architecture for the proposed meta-learner $g_\phi$

As mentioned in Section 3.3 in the main paper, the architecture of the proposed meta-learner $g_\phi$ is a 3-layer MLP. Each layer consists of $2N$ hidden units, where $N$ is the number of layers of the base learner network, $f_\theta$. This is because the meta-learner is conditioned on the layer-wise mean of gradients and weights of the base learner network at each inner-loop update step. ReLU activation function is placed between MLP layers.

## F  Visualization

In Section 4.5 of the main paper, the visualization of generated hyperparameters during meta-test is shown for only a bias term of a 4-convolutional layer. To further examine the dynamic behavior of our proposed adaptive update rule, the generated values across tasks, layers, and update steps for different initializations are plotted in Figure A, Figure B, and Figure C, respectively. There are several observations to make from the figures.

For different initializations, the ranges of generated values are different. This is especially evident for the generated learning rate $\alpha$, where the magnitude of values is diverse. This hints that each initialization prefers different learning dynamics, thus stressing the importance and effectiveness of ALFA. Furthermore, one should note the drastic changes in values across steps for each layer. In particular, this is prominent for the regularization term $\beta$ for different layers and the learning rate $\alpha$ for Conv3, Conv4, and linear bias, where the generated values change (up to the order of $1e{-}1$). Because the changes across inner-loop steps are so great, the variations across tasks are not visible. Thus, each plot includes a zoomed-in boxplot for one step due to the limited space. While not as dynamic as across steps, the variations across tasks are still present (up to the order of $1e{-}3$ for both $\alpha$ and $\beta$). This is still significant, considering how the usual inner-loop learning rate is from $1e{-}2$ to $1e{-}1$ and the usual $\ell_2$ weight decay term is in the order of $1e{-}6$ or $1e{-}5$, depending on the learning rate.

Overall, different dynamic changes across layers and initializations as well as variations across tasks and inner-loop steps further underline the significance of the adaptive learning update rule in gradient-based meta-learning frameworks.

Figure A: ALFA+Random Init: Visualization of the generated hyperparameters, $\boldsymbol{\alpha}$ and $\boldsymbol{\beta}$, across inner-loop steps and layers for a base learner of backbone 4-CONV. The proposed meta-learner was trained with a random initialization on 5-way 5-shot miniImagenet classification.

Figure B: ALFA+MAML [4]: Visualization of the generated hyperparameters, $\alpha$ and $\beta$, across inner-loop steps and layers for a base learner of backbone 4-CONV. The proposed meta-learner was trained with MAML initialization on 5-way 5-shot miniImagenet classification.

Figure C: ALFA+MAML+L2F [2]: Visualization of the generated hyperparameters, $\alpha$ and $\beta$, across inner-loop steps and layers for a base learner of backbone 4-CONV. The proposed meta-learner was trained with MAML+L2F initialization on 5-way 5-shot miniImagenet classification.