[Reviews · NeurIPS 2020]

Review 1

Summary and Contributions: Updated review: After reading the rebuttal, I believe this is a robust empirical paper and worth publishing for the following reasons: - The results are very strong and surprising, challenging emerging hypotheses about generalisation even for image datasets, which are very well studied. This paper convincingly shows that betting on meta-learning adaptation does indeed generalise better than packaging non-adaptive priors via pre-training. This was not at all clear before this paper; indeed, two reviewers needed some convincing to believe it. - The approach is efficient and scalable, contrary to some objections; the authors should indeed improve the text with the sentences provided in the rebuttal, and I believe they will. It's on them that such confusion arose in the first place. Same goes for the title. - The authors have taken our objections and questions very seriously: the rebuttal answers many of our questions and reinforces the already impressive empirical results. - Alpha+Random experiments convince me that the technical deltas to Meta-SGD and Meta-Opt are indeed significant. Priors needed for SOTA generalisation on these well studied datasets can be "packed" into the learning rule itself via meta-learning. It's very unlikely that prior works cited can support this claim on their own. - Although I do acknowledge I am curious about RL results, I am glad the authors chose to perform ablation and cross-dataset transfer experiments instead; I do not believe that missing RL results is itself grounds for rejection, especially since the classification results are surprising and need extensive ablations to be fully believed. I do welcome the extra regression experiments provided in the rebuttal, and I believe the revised draft will already be bursting with results and analyses. Old Review: The paper proposes a novel gradient-based meta-learning approach, nicknamed ALPHA, a fitting addition to the MAML zoo. The main contribution is to revive interest in the general class of gradient-based fast adaptation algorithms such as MAML, by providing excellent empirical results on standard benchmarks, as well as strong cross dataset generalization. However, the proposed method is largely orthogonal to MAML and the paper demonstrates virtuous combination of said algorithms. Analysis of meta-learned quantities rounds up the paper, together with an ablation of the main empirical result.

Strengths: The paper introduces a novel and generic meta-learning algorithm which modulates learning rates and L2 regularization constraints per adaptation step, per model layer and per task instance. This is done at the expense of a little over: [5(steps) + 1(first)] x 2 (alpha & beta) = 12 times the number of meta-learned parameters compared to MAML for the best empirical result; however, most memory is consumed by activations, not model parameters, so the memory overhead compared to MAML should be minimal. Computational cost should be on the same order as MAML, but I am left wondering what that means in practice. I have only cursorily read the code, so please let me know if my conclusions are not entirely right. As far as I know the reported miniImageNet and tieredImageNet results are state-of-the-art on the standard benchmark variants, without using any extra information. I find these results "too good to be true", but I have every reason to believe them, since the paper already answers most of my questions. I did not imagine such generalization is possible without pre-training the feature extractor. Personally, I was surprised that it is possible to generalize so well with ALPHA + Random. It seems to perform almost SOTA and is often stronger than MAML variants. I believe this is an equally strong contribution to the SOTA result. I was very surprised that such few samples could be used to tune millions of parameters from scratch in 5 steps of gradient descent on the same minibatch. The ablation study satisfies my curiosity w.r.t. several details of the main result. Good transfer to meta-test set of other datasets is particularly encouraging that performance is not due to some kind of fitting to dataset specifics, although CUB is sampled from the same larger dataset, but is indeed much more specific. I would like to see more similar experiments in future!

Weaknesses: Suggestions for improvement: - A discussion of computational costs compared to MAML would be interesting. How much slower is meta-training with ALPHA? - More emphasis could be put on the ALPHA+Random algorithm. Could you perhaps look at per task final adaptive parameter norms and compare ALPHA+Random, pure MAML and the main result? I suspect that all MAML variants substantially modify parameters based on the very biased training samples, while ALPHA relies heavily on model "prior" inductive bias. - Perhaps another set of boxplots in the appendix could be plotted with deviations from the initialization, if informative. Both learning rates and regularization coefficient matter in relation to the scale of gradients at each step, and aren't directly comparable if gradient scale falls dramatically. - I am curious how the authors rationalize and interpret negative learning rates, especially for variants which include MAML. - Should some L2 regularization be added, or be meta-learned, for the parameter initialization as well? Negative learning rates tell me that some amount of meta-overfitting may be due to the initialization being updated via MAML. It may turn out that optimal results are somewhere in between Random and MAML initializations, e.g. MAML with some (meta-learned) weight decay. - A discussion of possible limitations of proposed method would be interesting. Can the authors imagine conditions in which ALPHA would be detrimental or lead to divergence? Some failure cases mentioned in code comments could be brought into the main paper.

Correctness: As far as I can tell experimental methodology is correct and follows standard practice, rendering results directly comparable with existing literature.

Clarity: While the paper is generally clear, I believe the focus on demonstrating state-of-the-art performance marginalizes the exposition of the main method. After looking at the code I believe both alpha and beta have meta-learned, per step bias values which are multiplicatively updated by the generator. Why is this not reflected in the main text? Perhaps a cartoon figure could be used to explain what's going on, since learning rates and regularization constants are very seldom different per step, per task, etc. However, in this few-shot regime the effect on generalization is miraculous, which in my book deserves a figure.

Relation to Prior Work: A number of recent and previous SOTA methods are discussed and compared with. Perhaps Meta-SGD should be better emphasized as previous work. LEO also meta-learns learning rates in both latent and parameter spaces, but is different enough because it works on top of pre-trained representations. Perhaps more historical references should be added too; the novelty and efficient implementation of ALPHA remain. In fact, I am willing to improve my score if such aspects are fixed and the full method is described properly in the main text!

Reproducibility: Yes

Additional Feedback:


Review 2

Summary and Contributions: This paper proposes a modification to MAML in the form of a network that ingests the network weights and gradients for a given task and returns regularization and learning rate parameters. The authors show that the addition of this network improves the effectiveness of MAML; indeed, this approach enables a random initialization to outperform standard MAML in some cases. The authors evaluate the ALFA method on miniImageNet and tieredImageNet, as well as transfer from miniImageNet to CUB.

Strengths: Overall the method is reasonably interesting and appears to yield substantial performance improvements relative to the complexity of the method. The paper is fairly clearly written.

Weaknesses: There are several experiments that, if added, would substantially improve the paper. I would be willing to improve my score if some or all of the following were added: - Regression experiments. The authors only evaluate classification experiments. To claim broad improvements over MAML, the authors should also investigate their approach on a broad set of experiments as in the MAML paper. While the RL experiments are not necessary, even basic regression experiments would give further insight into the approach. - The authors should investigate 20-way classification, not just 5-way. - An ablation of performance providing only weights or only gradients to the hyperparameter selection network. To summarize, to convince a reader that the presented approach is a broadly applicable tool to improve MAML, the authors should investigate the method on a wide variety of problems. Finally, section 4.4.2 claims that there is a consistent performance improvement throughout the steps, whereas that does not appear to be a statistically significant result.

Correctness: The paper appears to be correct.

Clarity: The paper is written clearly.

Relation to Prior Work: This paper is an extension to a line of work proposing to augment the set of learnable parameters in MAML, with a particular focus on learning rates. This paper appears to be a simple and effective extension to these works, and positions itself correctly within this literature.

Reproducibility: Yes

Additional Feedback: The title of the paper is very vague. While I understand that the meta-learning literature is quite crowded and standing out is to some degree important, the title of the paper contains effectively no information. Post-Rebuttal: I thank the authors for including the additional experiments requested. I have increased my score. I would re-iterate that the title should be changed to make it more specific to the methods developed.


Review 3

Summary and Contributions: This paper proposes a method to obtain a better inner loop optimisation procedure for Model Agnostic Meta Learning by learning a hyperparameter network that generates per-layer learning rate and weight decay parameters for each inner loop adaptation step. The authors demonstrate that improving the inner loop adaptation process via adaptive per-layer learning rates and weight decay (adaptive based on the base learner's parameter values and loss gradients) can lead to effective few shot learning parameters with/without domain adaptation, even from a random initialisation.

Strengths: The paper has mostly clear empirical evaluations, considering some standard benchmark few shot learning tasks in the main paper and additional domains in the supplementary material. The claims made appear to be well supported by the experimental results -- even from a random initialisation, using the authors's update scheme ALFA leads to impressive performance. The idea of metalearning optimisation parameters is not entirely new in metalearning, (eg, seen in How To Train Your MAML, for inner loop learning rate schedules) but generating these conditioned on some state of the base network is a nice idea and has effective results. The Ablation studies are mostly clear, and provide additional insight.

Weaknesses: Recent few shot learning methods often evaluate on the Meta Dataset benchmark, which is larger scale and more thorough than the domains considered in this work. It would be nice to see results on this to more thoroughly assess the empirical claims of the authors. This is particularly the case because the CIFAR results in supplementary are less convincing than those on mini imagenet in the main paper. Update post author response: Thank you for including the additional results -- these help to strengthen the evaluation.

Correctness: There are particular aspects of this work that concern me that would benefit from author response. The biggest concern I have is that I find it challenging to understand how with only 25 examples total (5 way, 5 shot) and from random init, that learning a 5 step inner loop adaptive optimisation process can achieve such good test set performance. 5 inner loop steps with 25 total examples is very small, and therefore I am not confident in this result. Methods that learn a good embedding and then some sort of robust decision boundary via a convex solver do not necessarily do this well with so few examples. Given no data augmentation (as far as I can tell) and just weight decay regularisation, I am not sure about achieving this good performance with such limited data. In addition, given the input to the network g() that generates the LR and WD is only the mean weight value and gradient per layer, this information seems insufficient to be able to truly generate effective adaptive LR schedules. In Table 4, the claim is that the accuracy increases with more steps, but this is not what is shown. Update post author response: Thank you for the clarification -- the inclusion of per-parameter weight decay helps answer my question about random initialisation, and the additional results also help strengthen the paper.

Clarity: Paper is overall well written and clear. Figures and tables are for the most part clear and understandable also.

Relation to Prior Work: Prior work is discussed clearly and this work is well distinguished from that work.

Reproducibility: Yes

Additional Feedback: Overall, this is slightly below acceptance threshold because I find it challenging to understand how with such few examples and a random init, 5 inner loop steps can lead to such good performance (unprecedented). With more evidence to help convince the correctness of this, I will be more inclined to recommend this for acceptance. Update post author response: I am satisfied with the changes the authors have made, and will therefore be increasing my score to recommend acceptance.


Review 4

Summary and Contributions: Some of my concerns have been addressed in this rebuttal, therefore I would like to improve my final score. However, I still think the novelity is a bit limited, therefore I will give a borderline to this paper. This paper proposes a new weight update rule that greatly enhances the fast adaptation process. Specifically, it introduces a small meta-network that can adaptively generate per-step hyperparameters: learning rate and weight decay coefficients. The experimental results validate that the adaptive learning of inner-loop updates is actually the key missing ingredient that was often neglected in the existing few-shot learning approaches.

Strengths: Instead of trying to find the best initialization, this paper proposes a new Adaptive Learning strategy for Fast Adaptation, which enables more effective training with task-conditioned inner-loop updates from any given initialization

Weaknesses: 1. The idea of this paper is incremental, which tries to introduce adaptive learning rate and weight decay in the inner-loop optimization process of MAML, so as to help improve the fast adaptability of the original MAML algorithm. What's more, [2] shares the same idea with this work, in which its inner layer can jointly optimizes the initialization, update direction, and learning rates. Besides, the form of Eq.(4) is consistent with [1], and the meta learner in this paper directly uses a MLP to learn learning rate and weight decay, which lacks modeling of training dynamics, therefore limits its modeling description in the whole optimization process. [1] Ravi S, Larochelle H. Optimization as a model for few-shot learning. In ICLR, 2017. [2] Li Z, Zhou F, Chen F, et al. Meta-sgd: Learning to learn quickly for few-shot learning[J]. arXiv preprint arXiv:1707.09835, 2017. 2. The meta learner is difficult to generalize to large-scale network training. The input dimension of the learner is equal to twice the network parameters, which is difficult to generalize to different network structure learning. 3. The explanation is far fetched for the experiment of Cross-Domain Few-Shot Classification. The domain differences between meta training and meta test are large, but meta training does not contain this domain difference, and there is no clear guide to learn meta learners, so as to enable it adaptively deal with the differences of domain changes. It is reasonable for meta training to include this domain variation, and the meta learner should learn the ability to deal with this variation.

Correctness: Yes

Clarity: Yes

Relation to Prior Work: Not exactly

Reproducibility: Yes

Additional Feedback: See Weaknesses

[Author Response · NeurIPS 2020]

We thank all the reviewers for helpful feedback. We will do our best to answer the reviewers' questions and concerns.
Because we could not address all the issues due to the lack of space, we will try to include them in the final version.
**[R2, R4] Additional experiments:** As R2 and R4 suggested, we show additional experiments for Meta-Dataset (Table
A), 20-way classification for a 4-CONV base-learner on miniImageNet (Table B(a)), and regression on randomly
sampled sinusoids as in MAML [7] (Table B(b)). Consistent improvements over MAML across a broad set of
experiments clearly validate the generalization capability of our method. Following the suggestion of R2, we also
perform further ablation study on the input to the hyperparameter generator network, $g_\phi$ (Table B(c)). We will include
the experimental results with more detailed analysis in the final version of the paper.

Table A: Meta-Dataset test accuracy (%) of fo-MAML vs ALFA+fo-MAML (Ours) that are trained on ILSVRC-2012

| Model | ILSVRC | Omniglot | Aircraft | Birds | Textures | Quick Draw | Fungi | VGG Flower | Traffic signs | MSCOCO |
|---|---|---|---|---|---|---|---|---|---|---|
| fo-MAML [†] | 36.09±1.01 | 38.67±1.39 | 34.50±0.90 | 49.10±1.18 | 56.50±0.80 | 27.24±1.24 | 23.50±1.00 | 66.42±0.96 | 33.23±1.34 | 27.52±1.11 |
| Ours | **51.09±1.17** | **67.89±1.43** | **66.34±1.17** | **67.67±1.06** | **65.34±0.95** | **60.53±1.13** | **37.41±1.00** | **84.28±0.97** | **60.86±1.43** | **40.05±1.14** |

[†] Authors of Meta-Dataset [37] use fo-MAML to denote first-order MAML .

Table B: Additional experimental results and the ablation study for input variations

(a) 20-way classification

| Model | 1-shot (%) | 5-shot (%) |
|---|---|---|
| MAML | 15.21±0.36 | 18.23±0.39 |
| ALFA+MAML | **22.03±0.41** | **35.33±0.48** |

(b) MSE error on regression

| Model | 5 shots | 10 shots | 20 shots |
|---|---|---|---|
| MAML | 1.24±0.21 | 0.75±0.15 | 0.49±0.11 |
| ALFA+MAML | **0.92±0.19** | **0.62±0.16** | **0.32±0.06** |

(c) Ablation studies on $g_\phi$

| Input | 5-shot (%) |
|---|---|
| weight only | 68.47±0.46 |
| gradient only | 67.98±0.47 |
| weight + grad (ALFA) | **69.12±0.47** |

**[R1, R5] Differences to prior works:** While Meta-SGD [19] and LEO [30] meta-learn inner-loop learning rates, they
stay fixed throughout tasks and inner-loop steps during meta-test. In contrast, ALFA learns to generate hyperparameters
to adapt the weight-update rule to each task and each step. Although Ravi *et al.*[28] include these adaptive properties,
learning weight-update rules directly through their black-box implementation is very difficult, resulting in the limited
performance. Instead, ALFA specifies the form of weight-update rule to include the learning rate and weight decay
terms that are adaptively generated (Eq. 4), making it practically much more effective. This novel formulation allows
ALFA to strike a balance between weight-update with meta-learned but fixed learning rate [19, 30] and direct learning
of complex weight-update [28]. We will include clearer discussion with prior works in the updated version of the paper.
**[R1, R4] ALFA+Random:** Compared to other MAML variants, ALFA+Random requires meta-learning per-parameter
weight decay (L136-137) (or learning rate). Note that once the random initialization is given, it should stay fixed
throughout meta-train and meta-test altogether (Algorithm 1). Although this increases the total number of parameters
for saving the model, the number of *trainable* parameters stays the same as ALFA+MAML. Thus, one interpretation
why random initialization works could be that the form of prior knowledge has shifted from initialization (MAML) to
weight decay (or learning rate). We have double-checked the reproducibility of the results for random initialization and
are confident in the correctness. We will publicly release the code and trained models if our paper gets accepted.
**[R2, R4] Claims for Table 4:** Note that each column of Table 4 is a separately trained model with different number of
inner-loop update steps. Our intended claim was that ALFA+MAML consistently outperforms MAML with 5 steps
(Table 1), regardless of the number of steps, even with a single step. We will clarify our explanation in the final version.
**[R1] Computational costs:** Denoting the number of inner-loop steps as $s$ and the number of layers of $f_\theta$ as $N$, we
follow the practices by TADAM [24] to control the range of generated values for stable training, and use $2 \times s \times N$
additional parameters. Including the number of parameters of $g_\phi$, ALFA introduces $2sN + 12N^2$ more parameters
compared to MAML. For 5-shot 5-way classification with ResNet12 model and 5 inner-loop steps, the average inference
time for ALFA+MAML is 645ms per task, which is 3% slower than 628ms for MAML. However, training with ALFA
converges faster; ALFA+MAML only required 25 epochs, whereas MAML needed 90 epochs for full convergence.
**[R1] Interpretation for negative learning rates:** We agree with R1 that negative learning rates can regularize the
MAML initialization and reduce meta-overfitting, to which MAML is susceptible [12, 42]. In some sense, this has a
similar effect to L2F [2], which dynamically attenuates the initialization. Another possible interpretation could be that
negative learning rates prevent overfitting to the support set, since updating the weights with negative learning rates
prevents the model from further adaptation to support examples for generalization on unseen examples.
**[R5] Input dimension and generalization:** The input to $g_\phi$ is layerwise mean of gradients and weights of $f_\theta$, and
hence its dimension is $2N$, where $N$ is **the number of layers of** $f_\theta$ (L130-134). The total number of parameters for $g_\phi$
is $12N^2$ (L142), which is much less than the number of parameters in $f_\theta$ (2% for 4-CONV model). Thus, ALFA is easy
to generalize to different networks and involves less number of parameters than Meta-SGD [19] and Ravi *et al.* [28].
**[R5] Cross-domain explanation:** We agree with R5 that our claim in section 4.3.2 is a bit strong. However, as
discussed in [4], we believe that the adaptation to novel support examples plays a crucial role in cross-domain few-shot
classification when the domain gap between meta-train and meta-test is large (L209-213). Although ALFA is not
designed to explicitly learn the domain gap, its adaptive learning capability on new support samples can handle the
domain gap to some extent as demonstrated in Table 2. As R5 mentioned, we believe that including the domain variation
directly in meta-training can improve the performance, which is an interesting direction for further research. We will
address the discussions and reflect them in the final version.

[Meta-Review · NeurIPS 2020]

The reviewers generally agreed that this paper brings an important contribution to the NeurIPS community. The experiments are thorough. The results are quite strong, and were surprising to some reviewers. The approach also is scalable. There are multiple changes that we urge the authors to make for the camera ready version of the paper: - The title is far too broad, and not at all informative of the key ideas in the paper. The title should be revised such that it gets across that it is a meta-learning paper, and such that it is specific enough that the new title could not be used to also describe other existing papers. [For example, the current title could be used to describe really any gradient-based meta-learning paper] - The scalability of the approach was unclear from the text. In the camera ready, the authors should include some of the clarifications mentioned in the author response. - Of course, the new experiments in the author response should also be included in the revised paper. Also, note that the first-order MAML results in the author response are from an out-of-date version of the meta-dataset paper. The camera-ready paper should report the updated results from v4 of the meta-dataset arxiv paper (linked here: https://arxiv.org/abs/1903.03096). It should also include results from other top-performing methods, including this paper: https://arxiv.org/abs/2003.11539 - For the regression experiments in the rebuttal, the authors should also run a comparison that controls for the total number of parameters. Vanilla MAML often performs significantly better on this problem when using a larger architecture.